# A critical evaluation for validation of composite and unidimensional postoperative pain scales in horses

Paula Barreto da Rocha[1], Bernd Driessen[2], Sue M. McDonnell[2‡], Klaus Hopster[2‡], Laura Zarucco[3‡], Miguel Gozalo-Marcilla[4‡], Charlotte Hopster-Iversen[5‡], Pedro Henrique Esteves Trindade[6‡], Thamiris Kristine Gonzaga da Rocha[6‡], Marilda Onghero Taffarel[7‡], Bruna Bodini Alonso[8‡], Stijn Schauvliege[9‡], Stelio Pacca Loureiro Luna[6]*

1 Department of Surgical Specialties and Anesthesiology, Medical School, São Paulo State University (Unesp), Botucatu, São Paulo, Brazil, 2 Department of Clinical Studies, New Bolton Center, School of Veterinary Medicine, University of Pennsylvania, Kennett Square, Pennsylvania, United States of America, 3 Dipartimento di Scienze Veterinarie, Università degli Studi di Torino, Grugliasco, Italy, 4 The Royal (Dick) School of Veterinary Studies and the Roslin Institute, The University of Edinburgh, Edinburgh, Midlothian, United Kingdom, 5 Department of Veterinary Clinical Sciences, Section of Medicine and Surgery, Faculty of Health and Medical Sciences, University of Copenhagen, Frederiksberg, Denmark, 6 Department of Veterinary Surgery and Animal Reproduction, School of Veterinary Medicine and Animal Science, São Paulo State University (Unesp), Botucatu, São Paulo, Brazil, 7 Department of Veterinary Medicine, Maringá State University, Maringá, Paraná, Brazil, 8 Faculty of Animal Science and Food Engineering, Sao Paulo State University, Botucatu, Brazil, 9 Department of Anesthesiology and Domestic Animal Surgery, Faculty of Veterinary Medicine, Ghent University, Ghent, Belgium

☯ These authors contributed equally to this work.
‡ These authors also contributed equally to this work.
* stelio.pacca@unesp.br

**Data Availability Statement:** All relevant data are within the paper and its Supporting information files.

## Abstract

Proper pain therapy requires adequate pain assessment. This study evaluated the reliability and validity of the Unesp-Botucatu horse acute pain scale (UHAPS), the Orthopedic Composite Pain Scale (CPS) and unidimensional scales in horses admitted for orthopedic and soft tissue surgery. Forty-two horses were assessed and videotaped before surgery, up to 4 hours postoperatively, up to 3 hours after analgesic treatment, and 24 hours postoperatively (168 video clips). After six evaluators viewing each edited video clip twice in random order at a 20-day interval, they chose whether analgesia would be indicated and applied the Simple Descriptive, Numeric and Visual Analog scales, CPS, and UHAPS. For all evaluators, intra-observer reliability of UHAPS and CPS ranged from 0.70 to 0.97. Reproducibility was variable among the evaluators and ranged from poor to very good for all scales. Principal component analysis showed a weak association among 50% and 62% of the UHAPS and CPS items, respectively. Criterion validity based on Spearman correlation among all scales was above 0.67. Internal consistency was minimally acceptable (0.51–0.64). Item-total correlation was acceptable (0.3–0.7) for 50% and 38% of UHAPS and CPS items, respectively. UHAPS and CPS were specific (90% and 79% respectively), but both were not sensitive (43 and 38%, respectively). Construct validity (responsiveness) was confirmed for all scales because pain scores increased after surgery. The cut-off point for rescue analgesia was $\geq 5$ and $\geq 7$ for the UHAPS and CPS, respectively. All scales presented adequate repeatability,

**Funding:** The present work was carried out with funding support of the São Paulo Research Foundation (FAPESP) - [https://fapesp.br/en] (thematic project 2017/12815-0), for providing financial support for acquisition of veterinary equipment and materials for data collection in Brazil, CAPES (Coordination for the Improvement of Higher Education Personnel) - [https://www.gov.br/capes/pt-br] for funding PBR's Master of Science Scholarship (Process 168965) and Dorothy Russell Havemeyer Foundation [http://www.havemeyerfoundation.org; award # 2016 5-27134 to University of Pennsylvania Equine Behavior Program, Sue M McDonnell] and Narkovet Consulting®, LLC, USA [http://www.narkovet.com; Award #SR-NVC-USA 02-2018] for covering over-hours work of veterinary technician Jaime Miller in this study. The funders had no role in study design, data collection and analysis, decision to publish, or preparation of the manuscript.

**Competing interests:** Professor Bernd Driessen and Associate Professors Klaus Hopster and Laura Zarucco are shareholders of Narkovet Consulting®, LLC, which provided parts of the funding for the present study. However, Narkovet Consulting®, LLC did not in any way or form impact these authors' adherence to all PLOS ONE policies on sharing data and materials as detailed in the online guide for authors (http://journals.plos.org/plosone/s/competing-interests) or had any influence on data presentation and interpretation. All other authors declare that they have no conflicts of interest.

**Abbreviations:** AN, anesthesiologist; AUC, area under the curve; CPS, orthopedic composite pain scale; EI, equine internist; LI, lead investigator; M0, preoperative; M1, postoperative, before analgesic rescue; M2, postoperative, after analgesic rescue; M3, 24h after surgery; MA, data of grouped moments (M0 + M1 + M2 + M3); PCA, principal component analysis; RE, reference evaluator; ROC, receiver operating characteristic; VS, naive observer (veterinary student); VT, veterinary technician.

criterion validity, and partial responsiveness. Both composite scales showed poor association among items, minimally acceptable internal consistency, and weak sensitivity, indicating that they are suboptimal instruments for assessing postoperative pain. Both composite scales require further refinement with the exclusion of redundant or needless items and reduction of their maximum score applied to each item or should be replaced by other tools.

## Introduction

Assessing intensity and duration of nociception/pain in animals is one of the challenges veterinarians are faced with since animals, unlike humans, cannot verbally communicate whether they have an unpleasant or aversive sensory and emotional experience [1]. In the past decade, equine species have been in the center of attention with regard to developing improved pain monitoring tools [2]. However, as in any prey and flight animal, pain assessment in the horse is more complex than in some other domestic species because they have evolved to show little behavioral signs of discomfort whenever being approached or sensing a dangerous situation [3]. Previously employed nonspecific and unidimensional scales/scoring systems to assess pain in this species, such as visual analog, simple numeric, or simple descriptive scales, have major limitations. Shortcomings of those scoring methods include brief observation periods and the scoring system per se, particularly when assessing animals with supposedly mild to moderate pain [4–6]. Among those, the simple descriptive scale is presumably the least sensitive to identify discrete changes in pain expression [5]. Another limitation related to the evaluator is a lack of experience with the species and/or understanding of species-specific discomfort behaviors [5].

To improve pain evaluation in horses, more complex behavioral [7] or multidimensional scoring systems including the recording of physiological variables [8], have been proposed for specific conditions such as colic [6] and orthopedic trauma [8, 9]. More recently, also, horse grimace scales have been developed to supplement pain assessment [10, 11]. These scales have been tested in animals undergoing castration [10, 12] or suffering from musculoskeletal pain [13], acute pain after experimental noxious stimulation [14], somatic and visceral pain [15], colic [15, 16] or laminitis [17].

Intra- and inter-evaluator reliability and scoring system sensitivity and specificity have been determined for the Unesp-Botucatu composite horse pain scale (UHAPS) [12] and the behavior-based horse pain scales [7, 18] in animals experiencing acute pain. Still, the performance of a pain-scoring instrument in the clinical setting is a critical step in confirming its reliability, which assesses the ability of the instrument to produce similar results when used at different times by different individuals [19], as well as its effectiveness in accurately capturing manifestations of pain and discomfort in equine patients. So far, the efficacy of the composite orthopedic pain scale (CPS) and the Equine Utrecht University scale for facial assessment of pain (EQUUS-FAP) [9] and Equine Acute Abdominal Pain Scale (EAAPS) [18] have been evaluated in a clinical setting.

Multiple confounding factors limit the value of currently available pain scoring systems for use in clinical practice. Among those are the requirement for extended observation periods, the physical proximity of an evaluator to the animal, which may interfere with the animal's display of pain-related behaviors, interactions of the evaluator with the animal and physical recording of vital parameters (e.g. heart and respiratory rates and body temperature), and the effect of time of day, anesthesia, and analgesia affecting the spontaneous pain behaviors [20].

Furthermore, physical restraint of locomotion by for example bandages and casts, food intake restraint by muzzles, or limited continued visibility of the entire animal or its face (in case of horse grimace scale scoring), due to less than optimal light conditions in the stall, compromise the proper assessment of the animal and therefore scoring.

The primary objective of the present clinical study was to compare psychometric properties obtained with unidimensional (visual analog, simple numeric, and simple descriptive scales) and two composite pain scales (UHAPS and CPS) by evaluators with variable backgrounds in grading perioperative pain in horses. For this purpose, intra- and inter-observer reliability, criterion and construct validity, item-total correlation, internal consistency, responsiveness, sensitivity, specificity, and cut-off scores for the need of rescue analgesia were determined. We hypothesized that the two composite pain scales are more reliable and valid than unidimensional scoring systems for assessing perioperative pain under clinical practice conditions.

## Materials and methods

The study was approved by the School of Veterinary Medicine (Unesp) Animal Use and Ethics Committee (protocol 1228/2017) and by the University of Pennsylvania Animal Care and Use Committee (protocol 806321). In Ghent University, no ethics committee protocol was required because the animals did not undergo any procedures except videotaping. Owners, however, signed an authorization stating that they permitted collection of scientific data, both during the hospitalization and treatment of the animal at the hospital, and to use these data anonymously for scientific publications.

The horses enrolled in this study included equine patients from the veterinary teaching hospitals of the Schools of Veterinary Medicine at the University of Pennsylvania, USA, at Unesp-Botucatu, Brazil, and at Ghent University, Belgium. Client consent forms explaining the experimental protocol were obtained from each owner before enrolling an animal into the study. As this was a study performed in client-owned animals, all surgical procedures were performed upon the request of the owners. The anesthetic protocols and analgesic therapies used were those of routine clinical practice at each institution. The primary clinician solely defined the need for rescue analgesia without any interference by the investigators.

### Animals

Thirty male and twelve female horses of different breeds mean weight of 493 ± 87 kg (280–662 kg) and age ranging from 1 to 25 (8 ± 6.7) years were studied (S1 Table). For inclusion in the study, horses were required to be over one year of age, weigh more than 200 kg, and accept halter placement. Of a total of 59 horses initially scheduled for inclusion in the study, 17 were excluded due to discharge before 24 hours post-surgery, technical problems with filming, or difficulty with handling. All horses underwent a physical examination and, when necessary, laboratory testing based on their specific needs; however, no animals were excluded because of the results of these tests. The majority of horses were acclimated for six or more hours in the stall. The acclimation interval was less for those that were admitted to the hospital in the morning of the day of their surgical procedure.

### Protocol

Each horse was placed in an individual stall, without physical contact to other animals, although the stalls allowed visual, auditory, and olfactory contact with other hospital patients. Hay and water were available *ad libitum* to all horses, except during food and water withholding in the 4–12 hours prior to induction of anesthesia. Horses were kept in their stalls on straw or wood shaving bedding or on a concrete floor.

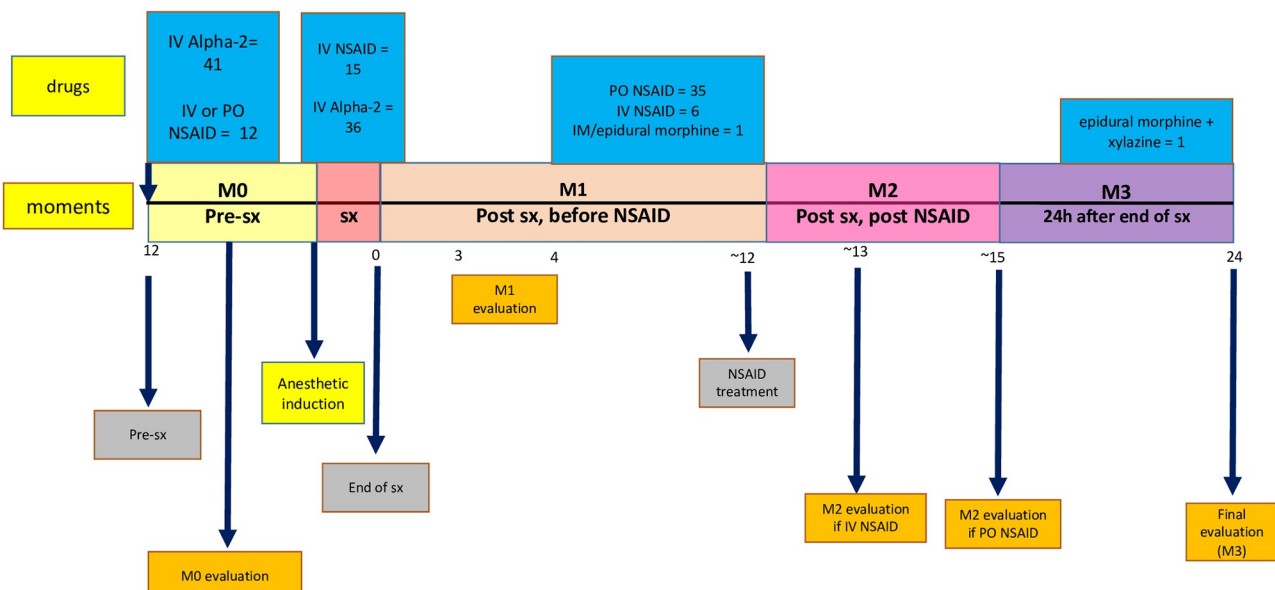

**Fig 1. Timeline of interventions and points at which drugs were given and pain ratings were performed.**

Anesthetic protocols varied based on the type of procedure performed (S2 Table). Patients were sedated with $\alpha_2$-agonists (xylazine, detomidine, dexmedetomidine, or romifidine). Acepromazine was also used in 22 horses as pre-anesthetic medication and perioperative opioids were used in 24 horses (butorphanol and/or morphine) (Fig 1). In 36 horses, non-steroidal anti-inflammatory drugs were administered IV or IM either in the preoperative or intraoperative periods. Anesthesia was induced with ketamine and midazolam or diazepam. Two horses also received guaifenesin. For maintenance of anesthesia isoflurane in 100% oxygen was used, except in three horses, in which anesthesia was maintained with desflurane in 100% oxygen. Two horses underwent procedures under sedation with continuous rate infusion of detomidine or romifidine. Thirty-two surgeries were performed in the morning and 10 in the afternoon. The responsible surgeon defined the need and timing of postoperative rescue analgesia, which always consisted of phenylbutazone 4.4 mg/kg BID (Butatabs E® or ButaJect®, Henry Schein Animal Health, Ohio–USA) orally (n = 34) or IV (n = 2), or flunixine meglumine 1.1 mg/kg BID IV (n = 6) (Injectable Flunixin UCB®, UCB Vet Saúde Animal, Jaboticabal- Brazil). Additionally, one horse received morphine 0.1 mg/kg via an epidural catheter (Morphine sulfate injection, USP, West-Ward ®, New Jersey, USA). Anesthetic recovery occurred in padded recovery rooms (n = 41) or a swimming pool (n = 1).

## Presential evaluation

Preoperative evaluation was performed between 1 and 12 hours before sedation (M0) and postoperative evaluations were performed up to 4 hours after the end of surgery (M1), up to 3 hours after IV or oral administration of analgesic drugs (M2), and 24 hours after the end of surgery (M3). The camera (GoPro Hero5 GoPro, Inc., California–USA) used to video record the evaluations was positioned outside the stall but in view of the horse. The recording was performed for 10 minutes without the presence of the evaluator (lead investigator), who remained at a distance from the stall, with no eye contact with the horse, and up to 5 minutes with the evaluator present, following the sequence of the CPS. Before entering the stall, the evaluator observed the horse's behavior for one minute and then entered the stall to measure

physiological variables (Fig 1). After the evaluation, the evaluator completed the scoring sheet in the following sequence: need or not for rescue analgesia according to the evaluator´s clinical experience (RA; being 0 = no and 1 = yes), simple descriptive (0 –no pain, 1 –mild pain, 2 – moderate pain, 3 –severe pain, 4 –very severe pain, 5 –worst possible pain), simple numeric (ordinal numbers form 0 –no pain to 10 –worst possible pain), visual analog scale (10 cm line from 0 –no pain to 10—worst possible pain), CPS [8] and the UHAPS [12]. Twenty of 32 horses at the University of Pennsylvania that were included in this report, simultaneously participated in another study that included uninterrupted 24-hour video recording for the duration of hospitalization [3].

## Remote evaluation

The video recordings were cut into sequences lasting between 2 and 4 minutes per moment by the lead investigator who produced clips that included examples of all behaviors displayed by the animals during the 15-minute recordings. The first 30 seconds consisted of displaying the horse without the presence of the evaluator in the stall and the remaining time included the presential appraisal period with the evaluator entering the stall, interacting with the horse, halter placement in cases where the horse was not already wearing a halter, free and guided locomotion, hay offering when allowed, and palpation of the surgery site(s). For remote evaluations, the video clips for the four time points (M0-M3) and for the horses were viewed in random order (Microsoft Excel® 2016). Six evaluators with varying degrees of equine clinical experience each independently viewed and evaluated the video clips in the order described before for on-site evaluation. These included one senior equine surgeon with many years of experience in assessing horses for pain and discomfort (later determined to be RE–reference evaluator), one anesthesiologist; one equine internist; the lead investigator who recorded and edited the video clips; an equine veterinary technician with over 20 years of clinical and research experience assessing discomfort and pain in horses; and a graduate student in veterinary medicine without clinical experience. These evaluators viewed each clip once (without any rewinding and reviewing), and immediately completed the scoring of the various scales in the same order as for the on-site evaluation. At least 20 days later, these evaluators repeated the procedure (second evaluation cycle). At the time of the first evaluation cycle, the evaluators had not been informed that they would be asked to repeat the evaluations. Physiological parameters that had been recorded during the on-site evaluation were used for remote scoring.

## Statistical analysis

Statistical analyses were performed as described in previous studies [12, 21–25] using R software in the Rstudio integrated development environment (Version 1.0.143 - ©2009–2016, Rstudio, Inc.). For all analyses, α was set to 5%. The rationale behind the psychometric tests performed in this study was based on premises described in Table 1. The methods are also described in Table 1.

   The traditional methodology for examining concurrent criterion validity is to correlate the scale under investigation against another instrument, ideally a gold standard [19]. For lack of any definitive fully validated scale that allows reliable and accurate pain assessment in horses that could serve as a reference or 'gold standard', an alternative approach for testing concurrent criterion validity was (1) to correlate the total scores of the composite scales with the simple numerical, simple descriptive and visual analog scales and (2) to compare all scales among each other scored by the reference evaluator in the 2nd evaluation phase. Another method used for concurrent criterion validation was the agreement between the reference evaluator and the

**Table 1. Methods for statistical validation of the Unesp-Botucatu horse acute pain scale (UHAPS), Composite Pain Scale (CPS), and unidimensional scales to assess postoperative pain in horses.**

| Type of analysis | Rationale | Description | Statistical test |
|---|---|---|---|
| **Multiple association**[*] | This analysis provides a general view of the instrument to establish the number of dimensions of the principal components and define the directions of each item eigenvector. It is expected that the vector of each item is directed to the time-points of pain. | **Principal component analysis**: analysis of the association of the items with each other to define the number of dimensions determined by different variables that establish the scale extension. | **Principal component analysis** ("princomp" and "get_pca_var" functions from the "stats" and "factoextra" packages respectively). According to the **Kaiser** criterion [26]. Significant association was considered when load values were $\geq 0.5$ or $\leq -0.5$, only in representative dimensions (eigenvalue $> 1$ and variance $> 20\%$) [27]. |
| **Intra-observer reliability**[**] | Assesses how close is the agreement between scores of the same measurement performed by the same individual on two different occasions to guarantee similarity of results. | **Repeatability**—the level of agreement of each observer with themselves was estimated by comparing the two phases of assessment, using the scores of each item, the total sum of the composite pain scales (CPS and UHAPS), simple numerical, simple descriptive, and visual analog scales as well as the need for rescue analgesia. | The **weighted Kappa coefficient** ($k_w$) was used to assess intra- and inter-observer reliability of **each item of the composite and total scores of unidimensional scales**, and the **need for rescue analgesia** (RA). Disagreements were weighted according to their distance to the square of perfect agreement. The confidence interval (CI) was estimated at 95%. $k_w$ ("cohen.kappa" function of the "psych" package). For the **visual analog scale**, the agreement-based **intra-class correlation coefficient (ICC)** type "agreement" and 95% CI were used since the data variability is higher when compared to the other unidimensional scales ("icc" function of the "irr" package) [28–30]. For the **total score** of the **composite scales**, the consistency type ICC and its 95% CI were used. Interpretation of $k_w$ and ICC: very good 0.81–1.0; good: 0.61–0.80; moderate: 0.41–0.60; reasonable: 0.21–0.4; poor $<0.2$. The $k_w$ and ICC$> 0.50$ were used as a criterion to refine the scale [30] |
| **Inter-observer reliability**[**] | Assesses how close the agreement is between scores of the same measurement performed by different individuals on the same occasion to guarantee similarity of results. | **1) Reproducibility**–the level of agreement between the reference evaluator and the other five observers was estimated, using the scores for each item; the total score of the composite and unidimensional scales, and the need for rescue analgesic. | |
| | | **2) Correlation matrix** was generated comparing the sum of scores of each observer versus all the other observers. | **Spearman rank correlation coefficient** ($r_s$; "rcorr" function of the "Hmisc" package). Interpretation of the degree of correlation $r_s$ good $\geq 0.75$; moderate: 0.5–0.74; poor $< 0.5$ [19]. |
| **Criterion validity**[*] | Correlation between the instrument and a gold standard measurement or a concrete outcome [19]. | **1) Concurrent validation test**: the correlation of the total scores of the composite scales were compared with the simple numerical, simple descriptive and visual analog scale scored by the Reference evaluator in the 2nd evaluation phase. | **Spearman rank correlation coefficient** ($r_s$; "rcorr" function of the "Hmisc" package). Interpretation of the degree of correlation $r_s$ good $\geq 0.75$; moderate: 0.5–0.74; poor $< 0.5$ [19]. |
| | | **2) Agreement** between the Reference evaluator and the other five evaluators (reproducibility) | Please see the **1) Reproducibility above** [21, 22] |
| **Item-total relation**[*] | Investigates if each item contributes to the total score of the scale in a homogeneous manner. If there is a great reduction in the correlation between each item and the total score of the scale when the item is removed, it means that the item has low correction and does not contribute to the total score of the scale. | **Homogeneity of the scale**. Identifies the relevance of each item on the scale and investigates inflationary items by correlating each item with the sum of all scale items, omitting that item. | **Spearman rank correlation coefficient** ($r_s$; "rcorr" function of the "Hmisc" package). Values between 0.3 and 0.7 were accepted [19]. |

(*Continued*)

**Table 1.** (Continued)

| Type of analysis | Rationale | Description | Statistical test |
|---|---|---|---|
| Construct validity* | **Responsiveness**: capacity of an instrument to determine a significant change of what it is supposed to measure (pain). It is expected that pain scores should increase, and horses should receive more rescue analgesia after surgery, compared to before surgery (see the cut-off point description at the last topic of this table below). | The scores of each item of the composite scales and the total scores of the composite and unidimensional scales, and the indication for rescue analgesia were compared over time among all time points. Interpretations assumed that the postoperative scores (M1 and M2) would be higher due to the likelihood of more intense pain in that period, while at M0 (pre-surgery) and M3 (24 hours), the score would be lower, with little or no pain, following the trend $M1 \approx M2 > M3 > M0$. <br><br> A second method to assess responsiveness was the comparison, by chi square, of the number of animals that would have a rescue analgesia score according to the post hoc determined Youden index [(Specificity + Sensitivity)– 1] at each time point, according to the assumption that scores would be as follows: $M1 \approx M2 > M3 > M0$. | For dichotomous variables (**need for rescue analgesic**) **logistic regression analysis** ("glm" function of the "stats" package) was applied using the **Tukey test** ("lsmeans" function of the "lsmeans" package) as a post hoc test. The independent variables were the moments of evaluation. Normal distribution of each variable at each time point was evaluated by box plots and histograms ("boxplot" and "histogram" functions of the "graphics" and "lattice" packages, respectively). **For non-parametric variables**, the **Friedman test** was used (function "friedman.test" of the package "stats"), with p-value corrected by the Bonferroni procedure (function "pairwiseSignTest" of the package "rcompanion") [21, 22] For **variables with normal distribution**, differences over time were evaluated by **ANOVA followed by the Tukey test**. |
| Internal consistency* | This analysis investigates the internal structure of the scale. Items meant to measure pain should generate homogenous results and should be interrelated. | Interrelation of the scores, which refers to the relationship of the scale items to each other. | **Cronbach's alpha coefficient** ($\alpha$; "cronbach" function of the "psy" package) Interpretation: 0.60–0.64, minimally acceptable; 0.65–0.69 acceptable; 0.70–0.74 good; 0.75–0.80 very good; and > 0.80 excellent [32, 33] |
| Sensitivity* | The true positive rate shown by the correct diagnosis of horses that are truly suffering pain. | The time period in which the highest postoperative scores (M2) were recorded for the composite scales was chosen to test sensitivity, thereby reflecting true positive results (horses suffering pain) [23]. Accordingly, for each domain M2 scores were transformed into dichotomous variables: 1 –the presence of pain referring to scores '1' or '2' or '3', and '0'–the absence of pain expression for a given item. For calculation of scores of sensitivity of the sum of scores at M2, the number of horses with a score of $\geq 5$ for the UHAPS and $\geq 7$ for the CPS scales was divided by the total number of horses [19], as the horses were expected to suffer from intense enough pain calling for rescue analgesia. | **S = TP/(TP+FN)** <br><br> **S** = sensitivity. **TP = true positive** (scores indicative of pain '1' or '2' or '3' at M2, calculated from horses with a total score of the UHAPS or CPS of $\geq 5$ and $\geq 7$, respectively,). **FN = false negative** (scores '0' at M2 in the same animals). Interpretation: excellent 95–100%; good 85–94.9%; moderate 70–84.9%; not sensitive <70%. Only items $\geq 70\%$ were included [8] |
| Specificity* | The true negative rate shown by the correct diagnosis of horses that are truly pain-free. | The time point in which the lowest postoperative scores (M0) were recorded for the composite scales was chosen to test specificity, thereby reflecting true negative results (pain free horses). The M0 scores were transformed into dichotomous items:1 –the presence of pain referring to scores '1' or '2' or '3', and '0'–the absence of pain expression for a given item. For calculation of specificity of the sum of scores at M0, the number of horses with a score of $< 5$ for the UHAPS and $< 7$ for the CPS scales were divided by the total number of horses [19] | **Sp = TN/(TN+FP)** <br><br> **Sp** = specificity. **TN = true negative** (scores indicating no expression of pain, i.e., '0', at M0, when the animal was expected to be pain-free, calculated with data recorded in horses that presented a total score of the UHAPS and CPS of $< 5$ and $< 7$, respectively). **FP = false positive** (scores indicating pain sensation, '1 or 2 or 3', at M0, when the same animals were expected to be pain-free). Interpretation: excellent 95–100%; good 85–94.9%; moderate 70–84.9%; not specific <70% [8] |

(*Continued*)

**Table 1.** (Continued)

| Type of analysis | Rationale | Description | Statistical test |
|---|---|---|---|
| **Rescue analgesic point**[*] | The score that lessens the possibility of errors (either false positive or negative results) is the Youden index. The area under the curve indicates the accuracy of the test. The AUC of an ideal test should be 1. If the AUC is 0.5, the test is the same as pure chance and is not distinguishing true positives from true negatives. The Youden index provides information of the cut-off point for the clinician whether to provide rescue analgesia or not. | Receiver operating characteristic (ROC) curve determined the minimum score calling for analgesic intervention. The **need of analgesia**, according to clinical experience was used as the true value and the **total score of the scales** as a predictive value to build a **ROC curve**. | **YI = (S+Sp) -1** |
| | | | **YI** = Youden Index; **S** = sensitivity; **Sp** = specificity. Analysis of the receiver operating characteristic curve (ROC; "roc" function of the "pROC" package) and the AUC: graphical representation of the relationship between the true positive (S) and the false positive (1-Sp). **Interpretation**: |
| | | Youden index [YI = (Sensitivity + Specificity)– 1)], determined the cut-off point for analgesic rescue [19, 21], representing the point of highest sensitivity and specificity at the same time [19, 21]. | Values can vary from 0 to 1: AUC = 1, perfect test, AUC 0.9–0.8, very good test, AUC 0.8–0.7, good test, AUC 0.7–0.6 sufficient test, AUC 0.6–0.5 bad test, while in a non-discriminatory test the AUC is less than 0 [35, 36]. |
| | | Area under the curve (AUC): indicates the discriminative ability of a test [34]. The closer the curve is to the upper left corner of the diagram, the better the discriminative ability of the test. | |

Software in the RStudio integrated development environment. For all analyzes, α was set to 5%, M0—preoperative; M1—postoperative, before analgesic rescue; M2—postoperative, after analgesic rescue; M3 - 24h after surgery; MA—data of grouped moments (M0 + M1 + M2 + M3).

[*] Data obtained from all moments grouped (MA) in the 2nd phase of the reference evaluator (RE) assessment;

[**] Data obtained from all moments grouped (MA) in the 2nd phase.

other five evaluators. These methods have been previously used for assessing pain scales in cats [22, 25], cattle [21], pigs [23], and sheep [24].

## Results

According to the confidence intervals of all evaluators, intra-observer reliability (repeatability) varied from good (minimal of 0.70) to very good (maximal value 0.97) in the sum of scores for both composite scales and ranged from moderate (minimal of 0.58) to very good (maximal of 0.99) for the unidimensional scales (Table 2). Repeatability based Kappa coefficient for each behavioral item of the UHAPS and CPS evaluated by the reference evaluator was very good for the majority of items (0.86–1) and only good for 'pawing' (0.77) and 'kicking the

**Table 2. Repeatability of the Unesp-Botucatu horse acute pain scale (UHAPS), Composite Pain Scale (CPS), and unidimensional scales to assess perioperative pain in horses.**

| | Intra-observer reliability [(kappa[*] or intraclass correlation[**] coefficients (confidence interval 95%)] | | | | | |
|---|---|---|---|---|---|---|
| | **Reference Evaluator** | **Lead Investigator)** | **Anesthesiologist** | **Veterinary Technician** | **Equine internist** | **Veterinary Student** |
| **UHAPS**[**] | 0.96 (0.94–0.97) | 0.95 (0.93–0.96) | 0.82 (0.76–0.87) | 0.80 (0.73–0.85) | 0.77 (0.70–0.83) | 0.79 (0.73–0.84) |
| **CPS**[**] | 0.95 (0.93–0.96) | 0.94 (0.91–0.95) | 0.83 (0.77–0.87) | 0.85 (0.80–0.89) | 0.80 (0.74–0.85) | 0.83 (0.78–0.87) |
| **Rescue analgesia**[*] | 0.92 (0.84–1.00) | 0.90 (0.81–0.98) | 0.74 (0.55–0.92) | 0.66 (0.52–0.81) | 0.67 (0.55–0.78) | 0.58 (0.44–0.72) |
| **Simple descriptive scale**[*] | 0.95 (0.93–0.98) | 0.94 (0.92–0.97) | 0.87 (0.81–0.92) | 0.83 (0.76–0.89) | 0.84 (0.79–0.90) | 0.72 (0.59–0.86) |
| **Simple numeric scale**[*] | 0.98 (0.97–0.99) | 0.98 (0.97–0.99) | 0.94 (0.91–0.96) | 0.82 (0.72–0.93) | 0.87 (0.82–0.93) | 0.70 (0.58–0.83) |
| **Visual analog scale**[**] | 0.98 (0.97–0.99) | 0.98 (0.97–0.98) | 0.94 (0.92–0.96) | 0.86 (0.82–0.90) | 0.88 (0.83–0.91) | 0.75 (0.67–0.81) |

Interpretation: 0.81–1.0 very good; 0.61–0.80 good; 0.41–0.60 moderate; 0.21–0.4 reasonable; < 0.2 poor [27, 31]

**Table 3. Observer matrix correlation for the Unesp-Botucatu horse acute pain scale (UHAPS) and the Composite Pain Scale (CPS) to assess perioperative pain in horses.**

|  | LI | | EI | | VT | | RE | | AN | |
|---|---|---|---|---|---|---|---|---|---|---|
|  | **UHAPS** | **CPS** | **UHAPS** | **CPS** | **UHAPS** | **CPS** | **UHAPS** | **CPS** | **UHAPS** | **CPS** |
| **Lead Investigator (LI)** | | | | | | | | | | |
| **Equine Internist (EI)** | 0.63 | 0.66 | | | | | | | | |
| **Veterinary Technician (VT)** | 0.61 | 0.63 | 0.46 | 0.59 | | | | | | |
| **Reference Evaluator (RE)** | 0.84 | 0.97 | 0.61 | 0.66 | 0.64 | 0.62 | | | | |
| **Anesthesiologist (AN)** | 0.65 | 0.66 | 0.55 | 0.56 | 0.64 | 0.67 | 0.68 | 0.67 | | |
| **Veterinary Student (VS)** | 0.53 | 0.47 | 0.39 | 0.47 | 0.56 | 0.52 | 0.54 | 0.47 | 0.71 | 0.6 |

Spearman's correlation coefficient: Good $\geq$ 0.75; moderate: 0.5–0.74; poor < 0.5 [19]. Physiological parameters were not included. P values were less than $<1.32^{-7}$ in all cases.

abdomen' (0.66) for each respective scale (S3 Table). The matrix Spearman inter-observer correlation was poor for the veterinary technician (0.46) and student (0.39) *vs* the equine internist for the UHAPS and veterinary student *vs* the other three evaluators for the CPS (0.47). Other correlations were moderate (0.53–0.71) or good (reference evaluator *vs* lead investigator; 0.84 and 0.97 for UHAPS and CPS respectively) in 80% of the comparisons (Table 3). The reference evaluator with the best repeatability among the other evaluators for both composite scales was the same that presented the best inter-observer correlations (Table 3).

Reproducibility combining phases 1 and 2 of video analysis based on inter-observer reliability comparisons against the reference evaluator was reasonable in 10%, moderate in 20%, good in 35% and very good in 35% of comparisons, for both composite scales (Table 4). Reproducibility ranged from poor to very good when each behavior was evaluated separately (S4 Table) and for the indication of rescue analgesia according to clinical experience, simple descriptive,

**Table 4. Reproducibility of the Unesp-Botucatu horse acute pain scale (UHAPS), Composite Pain Scale (CPS), and unidimensional scales to assess perioperative pain in horses.**

| Inter-observer reliability [(kappa* or intraclass correlation** coefficients (confidence interval 95%)] | | | | | | |
|---|---|---|---|---|---|---|
| | | Reference evaluator *versus* | | | | |
| | | Lead investigator | Anesthesiologist | Veterinary technician | Equine internist | Veterinary student |
| **UHAPS**** | **Phase 1** | 0.85 (0.80–0.89) | 0.74 (0.66–0.80) | 0.67 (0.57–0.74) | 0.49 (0.37–0.60) | 0.48 (0.35–0.59) |
| | **Phase 2** | 0.83 (0.78–0.87) | 0.73 (0.65–0.79) | 0.63 (0.53–0.71) | 0.40 (0.26–0.52) | 0.49 (0.36–0.59) |
| **CPS**** | **Phase 1** | 0.97 (0.96–0.98) | 0.79 (0.72–0.84) | 0.65 (0.56–0.73) | 0.64 (0.54–0.72 | 0.63 (0.53–0.71) |
| | **Phase 2** | 0.96 (0.94–0.97) | 0.63 (0.53–0.71) | 0.70 (0.62–0.77) | 0.33 (0.19–0.46) | 0.43 (0.30–0.55) |
| **Rescue analgesia indication*** | **Phase 1** | 0.85 (0.75–0.95) | 0.6 (0.44–0.75) | 0.22 (0.13–0.30) | 0.56 (0.38–0.73) | 0.36 (0.21–0.52) |
| | **Phase 2** | 0.78 (0.66–0.89) | 0.23 (0.14–0.32) | 0.47 (0.28–0.65) | 0.59 (0.41–0.77) | 0.53 (0.37–0.68) |
| **Simple descriptive scale*** | **Phase 1** | 0.94 (0.92–0.95) | 0.73 (0.66–0.80) | 0.64 (0.54–0.72) | 0.75 (0.67–0.81) | 0.23 (0.08–0.36) |
| | **Phase 2** | 0.93 (0.91–0.96) | 0.77 (0.68–0.87) | 0.59 (0.49–0.69) | 0.76 (0.69–0.82) | 0.37 (0.25–0.49) |
| **Simple numeric scale*** | **Phase 1** | 0.98 (0.97–0.98) | 0.79 (0.72–0.84) | 0.67 (0.58–0.74) | 0.82 (0.77–0.87) | 0.22 (0.07–0.35) |
| | **Phase 2** | 0.98 (0.97–0.99) | 0.82 (0.73–0.90) | 0.66 (0.54–0.77) | 0.84 (0.78–0.89) | 0.33 (0.22–0.45) |
| **Visual analog scale**** | **Phase 1** | 0.98 (0.97–0.98) | 0.75 (0.68–0.81) | 0.68 (0.59–0.75) | 0.78 (0.72–0.84) | 0.17 (0.02–0.32) |
| | **Phase 2** | 0.98 (0.97–0.98) | 0.83 (0.77–0.87) | 0.64 (0.54–0.72) | 0.79 (0.72–0.84) | 0.22 (0.07–0.36) |

Interpretation: 0,81–1.0 very good; 0.61–0.80 good; 0.41–0.60 moderate; 0.21–0.4 reasonable; < 0.2 poor [31]. Reference evaluator was the one with the highest intra-observer reliability.

**Table 5. Load, eigenvalue, and variance of the Unesp-Botucatu horse acute pain scale (UHAPS) and the Composite Pain Scale (CPS) submitted to principal component analysis.**

| UHAPS | Dimension 1 | CPS | Dimension 1 |
|---|---|---|---|
| Positioning in the stall | **-0.60** | Appearance | **0.72** |
| Locomotion | **-0.63** | Kicking at the abdomen | 0.29 |
| Locomotion when led by the evaluator | **-0.68** | Pawing on the floor | **0.64** |
| Response to palpation of the painful area | **-0.54** | Posture | **0.75** |
| Looking at the flank | -0.22 | Head movement | **0.75** |
| Kicking at the abdomen | -0.02 | Appetite | 0.34 |
| Lifting hind limbs | -0.45 | Response to observer | 0.46 |
| Head movements | **-0.62** | Response to palpation of the painful area | 0.37 |
| Pawing on the floor | -0.29 | HR | 0.34 |
| HR | -0.31 | RR | 0.05 |
| | | Digestive | 0.29 |
| | | Temperature | 0.40 |
| | | Sweating | -0.15 |
| **Eigenvalue** | 2.34 | **Eigenvalue** | 2.96 |
| **Variance** | 23.4% | **Variance** | 23.0% |

Bold values indicate items with a load value $\geq 0.50$ or $\leq$ -0.50 indicating an association amongst each other Eigenvalue > 1 and variance >20% [26]

simple numerical and visual analog scales (Table 4). Reproducibility changed little when data from phase 1 were compared to phase 2.

Only one dimension of the principal component analysis of the CPS and UHAPS revealed a representative eigenvalue and variance. Therefore, both scales are unidimensional. Only 31% of the CPS items and 50% of the UHAPS items showed a significant association (Table 5).

In the criterion analysis, when comparing the UHAPS and CPS with the simple descriptive, simple numeric, and visual analog scales, the Spearman correlation was between 0.72 and 0.77, and 0.67 between the composite scales (Table 6), demonstrating criterion validity.

When pooled data, i.e. ratings at all four-time points were considered, for the UHAPS, item-total correlation was below the acceptable value ($< 0.3$) for 'looking at the flank', 'kicking at the abdomen', 'lifting hind limbs', 'pawing on the floor' and heart rate (50% of the items—Table 7). For CPS scale, item-total correlation was below the acceptable value ($< 0.3$) for the items kicking at the abdomen', 'response to the observer' and 'to palpation', heart and respiratory rates, 'digestive', and 'temperature', therefore 38% of the items were below the acceptable value (Table 8). The practical information of these results is that the items that are below the acceptable correlation value are not contributing to the entire scale.

**Table 6. Criterion validity based on the correlation between the Unesp-Botucatu horse acute pain scale (UHAPS) and the Composite Pain Scale (CPS) *vs* unidimensional scales to assess perioperative pain.**

| | UHAPS | CPS |
|---|---|---|
| **Simple descriptive scale** | 0.74 | 0.72 |
| **Simple numeric scale** | 0.77 | 0.74 |
| **Visual analog scale** | 0.76 | 0.72 |
| **CPS** | 0.67 | - |

Interpretation of Spearman's correlation coefficient: Good $\geq 0.75$; moderate: 0.5–0.74; poor $< 0.5$ [19] (p < 0.001 in all cases). P values were less than $<2.2^{-16}$ in all cases.

**Table 7. Item-total correlation and internal consistency, specificity, and sensitivity for the Unesp-Botucatu horse acute pain scale (UHAPS).**

| Item | Item-total correlation (Spearman) | Internal consistency (Cronbach's α) | Specificity (%) | Sensitivity (%) |
|---|---|---|---|---|
| Full scale | | 0.60 | 90 | 43 |
| | *Excluding each item below* | | | |
| Positioning in the stall | **0.39** | 0.55 | 86 | 38 |
| Locomotion | **0.39** | 0.55 | 79 | 40 |
| Locomotion when led by the evaluator | **0.44** | 0.54 | 90 | 33 |
| Response to palpation of the painful area | **0.36** | 0.56 | 90 | 40 |
| Looking at the flank | 0.13 | 0.60 | 93 | 5 |
| Kicking at the abdomen | 0.02 | 0.61 | 100 | 0 |
| Lifting hind limbs | 0.27 | 0.59 | 100 | 10 |
| Head movements | **0.56** | 0.54 | 88 | 36 |
| Pawing on the floor | 0.15 | 0.60 | 98 | 10 |
| Heart rate | 0.19 | 0.61 | 100 | 10 |

*Item-total correlation*—interpretation of Spearman's coefficient: Values between 0.3 and 0.7 were considered acceptable [19]. *Internal consistency*: Cronbach's α coefficient based on item correlation was calculated for the overall score and by excluding each item of the scale. Values for the α coefficient were categorized as follows: 0.60–0.64, minimally acceptable; 0.65–0.69, acceptable; 0.70–0.74, good; 0.75–0.80, very good; and > 0.80, excellent [37]. *Specificity*: The M0 scores were transformed into dichotomous items:1 –the presence of pain referring to scores '1' or '2', and '0'–the absence of pain expression for a given item. For calculation of specificity at M0, only horses with a score of < 5 for the UHAPS and < 7 for the CPS scales were used [19]. Results were classified as excellent (95–100%), good (85–94.9%), moderate (70–84.9%), or non-specific (<70) [8]. *Sensitivity*: M2 scores were transformed into dichotomous variables: 1 –the presence of pain referring to scores '1' or '2', and '0'– the absence of pain expression for a given item. For calculation of the sensitivity of the sum of scores at M2, only horses with a score of ≥ 5 for the UHAPS and ≥ 7 for the CPS scales were used at M2. Results were classified as excellent (95–100%), good (85–94.9%), moderate (70–84.9%), or non-sensitive (<70) [8].

Both UHAPS and CPS presented minimally acceptable internal consistency (Cronbach's α coefficient ≈ 0.60) (Tables 7 and 8). Internal consistency defines how the items mutually influence each other. If the target item is well fit with the other items, one would expect that, after removing it, internal consistency would reduce compared to the internal consistency of the full scale. Both for UHAPS and CPS, the same items not approved by item-total correlation (not in bold) were the ones that, when removed, either affected very little or even increased internal consistency results, indicating that they do not contribute significantly to the scale, because when they were removed, internal consistency was improved.

With the exception of 'appetite' in CPS, for both scales, the same items showing low load values in the principal component analysis were the ones that both were not approved by item-total correlation and showed weak internal consistency.

For the UHAPS, the specificity ranged from 79–100% (Table 7) and for the CPS, specificity ranged from 69% (respiratory rate) to 100%. (Table 8). Neither one of the scoring systems showed proper sensitivity (Tables 7 and 8). Low sensitivity results indicate that none of the scales were good enough to recognize pain in horses that were possibly suffering pain (true positives), otherwise the UHAPS and CPS correctly diagnosed pain-free horses in 90% and 79% of the cases, respectively (specificity).

With regard to responsiveness, the scores given by all evaluators were greater at M1 (post-surgery, before analgesic rescue) and for four evaluators at M2 (after analgesic rescue) when compared to the baseline (M0) for the UHAPS (Fig 2; Table 9; S5 Table). Except for the veterinary student, the same result occurred for the unidimensional scales (Table 9; S5 Table). CPS scores were greater at M2 than at M0 only for the reference evaluator and lead investigator, and greater at M1 than M0 only for the anesthesiologist and veterinary technician (Fig 3;

**Table 8. Item-total correlation and internal consistency, specificity, and sensitivity for composite pain scale.**

| Item | Item-total correlation (Spearman) | Internal consistency (Cronbach's α) | Specificity (%) | Sensitivity (%) |
|---|---|---|---|---|
| Full scale | | 0.59 | 79 | 38 |
| | *Excluding each item below* | | | |
| Appearance | **0.43** | 0.51 | 86 | 29 |
| Kicking at the abdomen | 0.19 | 0.58 | 100 | 7 |
| Pawing on the floor | **0.33** | 0.56 | 98 | 10 |
| Posture | **0.43** | 0.51 | 88 | 31 |
| Head movement | **0.53** | 0.52 | 93 | 31 |
| Appetite | **0.42** | 0.59 | 79 | 33 |
| Response to observer | 0.25 | 0.57 | 100 | 2 |
| Response to palpation of the painful area | 0.27 | 0.57 | 81 | 31 |
| Heart rate | 0.19 | 0.58 | 93 | 7 |
| Respiratory rate | 0.03 | 0.64 | 69 | 21 |
| Digestive | 0.18 | 0.58 | 90 | 21 |
| Temperature | 0.08 | 0.58 | 95 | 7 |
| Sweating | -0.11 | 0.62 | 98 | 2 |

*Item-total correlation*—interpretation of Spearman's coefficient: Values between 0.3 and 0.7 were considered acceptable [19]. *Internal consistency*: Cronbach's α coefficient based on item correlation was calculated for the overall score and by excluding each item of the scale. Values for the α coefficient were categorized as follows: 0.60–0.64, minimally acceptable; 0.65–0.69, acceptable; 0.70–0.74, good; 0.75–0.80, very good; and > 0.80, excellent [32, 33, 37]. *Specificity*: Number of horses with scores 0 for a given item at M0 divided by the total number of horses at M0. For calculation of specificity of the sum of scores at M0, according to the Youden index (see Table 10), the number of horses with a score of < 5 for the UHAPS and < 7 for the CPS scales were divided by the total number of horses [19] [*Sensitivity*: Number of horses at M2 with scores '1' or '2' or '3' for a given item divided by the total number of horses. For calculation of sensitivity of the sum of scores at M2, the number of horses with a score of ≥ 5 for the UHAPS and ≥ 7 for the CPS scales was divided by the total number of horses [19] (see Table 1). Results were classified as excellent (95–100%), good (85–94.9%), moderate (70–84.9%), or non-specific or sensitive (<70) [8].

Table 9; S5 Table). Indication for the need of rescue analgesia based on the clinical experience-driven judgment was greater at M1 and M2 vs M0, only according to the veterinary student (S5 Table).

The distribution of scores for each item is presented for each time point in Figs 4 and 5 for the UHAPS and CPS, respectively. This information assesses the relevance of each graduation within the item. 'Position in the stall' was the only item where score 2 appeared to be relevant for the UHAPS. 'Appetite' and 'respiratory rate' were the only items where score 3 were relevant in the CPS.

The cut-off points for rescue analgesia were defined by the receiver operating characteristic (ROC) curve and the Youden index. They were ≥ 5 for the UHAPS (Area under the curve—AUC 0.89), ≥ 7 for the CPS (AUC 0.90), ≥ 6 for visual analog (AUC 0.99), ≥ 3 of 5 for simple descriptive (AUC 0.98) and ≥ 5 of 10 for simple numeric scales (AUC 0.99) (Fig 6; Table 10). When these scales are used to assess pain, these scores suggest that rescue analgesia is indicated, as they represent the optimal simultaneous result of sensitivity and specificity, minimizing the possibility of providing analgesia to a pain-free horse and maximizing the possibility of providing analgesia to a horse suffering pain.

The number of horses presenting reference evaluator pain scores equal to or above the Youden Index increased progressively from M0 to M2 and decreased at M3 (24h after surgery) in both composite scales (Table 11). Differences were observed between M0 and M2 for the UHAPS only (Chi-square $X^2$ 10.41, p 0.00). Indication of rescue analgesia based on

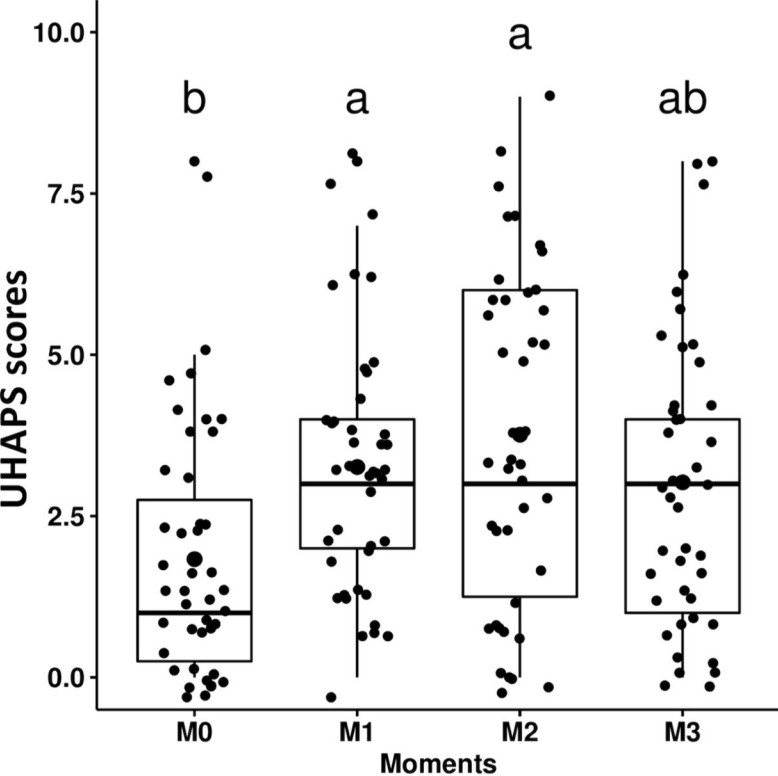

**Fig 2. Box plots of the total scores of the Unesp-Botucatu horse acute pain scale (UHAPS) over time.** M0 –before surgery; M1 –up to 4 hours after surgery; M2—up to 3 hours after analgesic treatment; M3–24 hours after surgery. Small letters represent statistically significant differences (a > b).

clinical experience-driven judgment and based on the Youden index of unidimensional scales was similar in all moments (Table 11). At M2 number of horses demanding rescue analgesia according to composite scales was higher than clinical experience and unidimensional scales.

**Table 9. Median (range) scores of the Unesp-Botucatu horse acute pain scale (UHAPS), the Composite Pain Scale (CPS), and unidimensional scales in the perioperative period in horses.**

|  | M0 | M1 | M2 | M3 |
|---|---|---|---|---|
| **UHAPS** | $1^b$ (0–8) | $3^a$ (0–8) | $3^a$ (0–9) | $3^{ab}$ (0–8) |
| **CPS** | $3^b$ (0–10) | $5^{ab}$ (0–15) | $5.5^a$ (0–18) | $4^{ab}$ (0–17) |
| **Rescue indication**[*] | 0 (0–1) | 1 (0–1) | 1 (0–1) | 1 (0–1) |
| **Simple descriptive scale** | $0^b$ (0–4) | $1^a$ (0–4) | $1^a$ (0–5) | $1^{ab}$ (0–5) |
| **Simple numeric scale** | $0^b$ (0–8) | $2.5^a$ (0–9) | $2^a$ (0–9) | $1^{ab}$ (0–10) |
| **Visual analog scale** | $0^b$ (0–8.8) | $2.7^a$ (0–10) | $2.1^a$ (0–10) | $1.1^a$ (0–10) |

Visual analog scale. Total scores of the reference evaluator were compared over time (M0 *vs* M1 *vs* M2 *vs* M3) by Friedman test, with p-value corrected by Bonferroni.

[*]For the dichotomic item 'rescue indication' (0 = no indication of rescue analgesia; 1 = indication of rescue analgesia), the logistical regression using the Tukey test as a post hoc test was applied.

M0 –before surgery; M1 –up to 4 hours after surgery; M2—up to 3 hours after analgesic treatment; M3–24 hours after surgery.

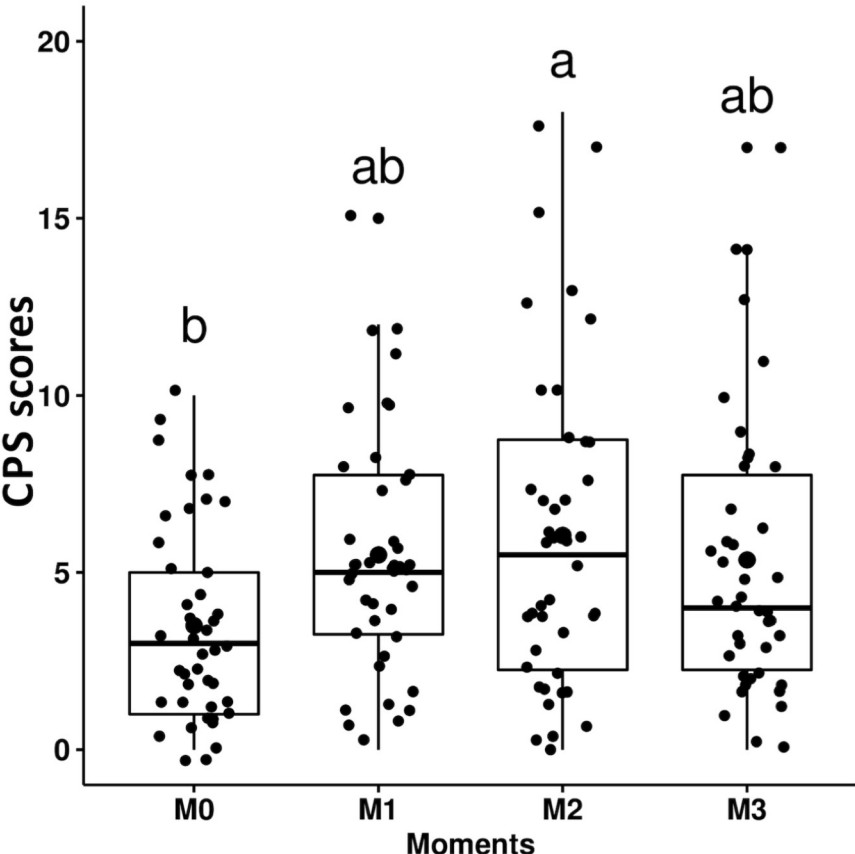

**Fig 3. Box plots of the total scores of the Composite Pain Scale (CPS) over time.** M0 –before surgery; M1 –up to 4 hours after surgery; M2—up to 3 hours after analgesic treatment; M3–24 hours after surgery. Small letters represent statistically significant differences (a > b).

## Discussion

The current study tested the UHAPS, CPS and unidimensional scales for assessment of perioperative pain in a clinical population of horses undergoing a wide variety of surgical procedures. To investigate whether observers with different backgrounds and skills could use the pain scoring instruments well, a rather diverse group of evaluators was picked, including veterinarians with different levels of experience in assessing horses in pain, a technician with extensive familiarity with behavioral assessment of horses and a student as an inexperienced (naive) observer. In addition, the contribution of each evaluation item to the total score was conducted, thereby identifying its relevance within each scoring system.

While the repeatability of the total scores for both composite scales ranged from good to very good and was similar to that observed in the original studies presenting and evaluating the scales [8, 12], the reproducibility of the composite scales was rather variable. Most of the evaluators´ correlations against the reference evaluator were good to very good, however, except for the naive observer, our original hypothesis that the two composite scoring systems would offer the advantage of better reliability over the unidimensional scales, must be rejected. This is a somewhat surprising result because in other species, composite behavioral scales perform better than unidimensional ones. The ability to distinguish different levels of pain/discomfort in rats was better with a behavioral *vs* a visual analog scale [38], and this was unrelated

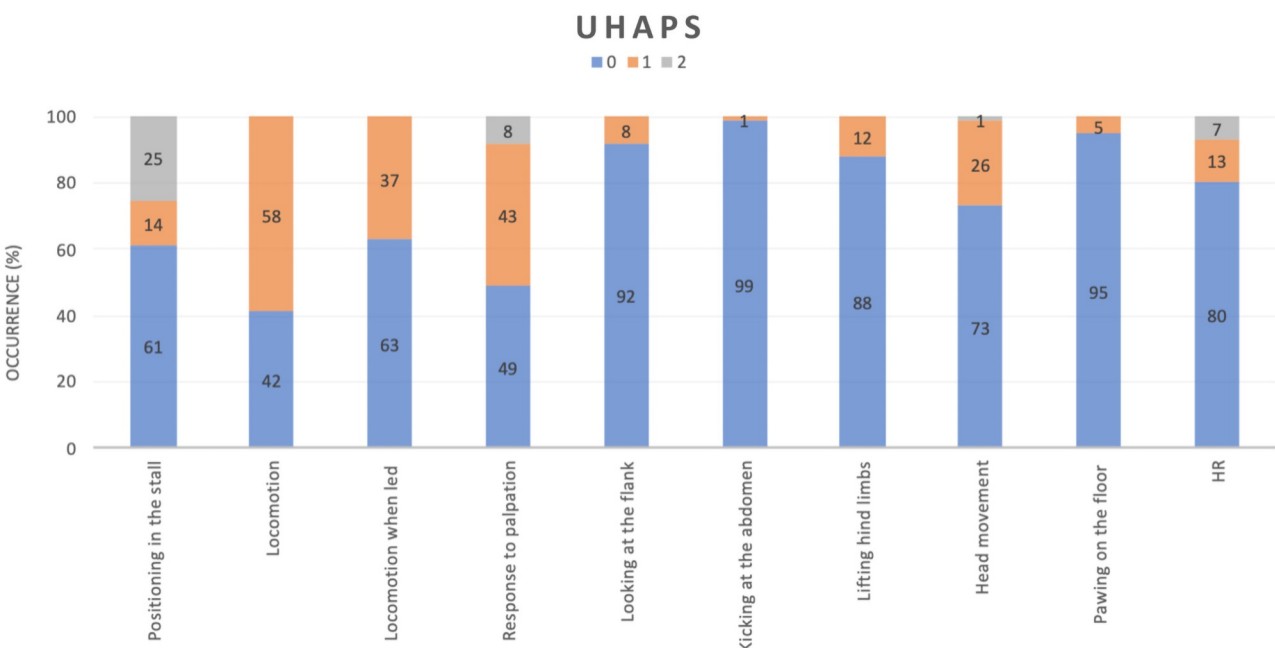

**Fig 4. Distribution of scores for each item of the Unesp-Botucatu horse acute pain scale (UHAPS) at all perioperative moments.**

to previous evaluators' experiences in laboratory animal management and behavior. In dogs, reproducibility of visual analog scale scoring of pain/discomfort was poor or moderate among anesthetists [39]. A very good agreement in pain scoring was found only between the reference evaluator and lead investigator, similar to previously reported data for the CPS scale, even when inexperienced persons (students) were the evaluators [9]. However, the raters in those previous studies were not blinded, as much as the lead investigator in the present study, who was aware of the case management, had performed on-site evaluations and video-recorded most of the equine patients. In contrast, the other five evaluators, for which inter-observer reliability was lower, were blinded. Therefore, the high inter-observer reliability for the lead investigator was likely due to observer bias, a commonly noted confounding factor in animal behavior research [40].

The wide variation in reproducibility discovered in this study for all scales requires critical analysis. Only differences in familiarity with the species in general and pain assessment in particular or with the pain scales tested do not explain the highly variable inter-observer reliability data. At the time of the study, except the veterinary student, all evaluators had 15–25 years of experience in clinical practice and thus with assessment of discomfort and pain in equine patients, yet their pain scorings varied a lot, independent of the scale. More experience obtained after phase 1 analysis did not improve reproducibility, even for the naive observer. Training evaluators in the use of the scoring systems and providing them with more detailed instructions and/or video demonstrations for each item of those scales, as is available online at www.animalpain.com.br for pain scales for cats [22], cattle [21], sheep [24] and pigs [23], may improve intra- and inter-observer reliability, even with experienced evaluators [38].

It is noteworthy that the need for rescue analgesia according to clinical experience had the lowest intra-observer reliability and a considerable variation on inter-observer reliability. This assessment depends on many factors other than the perceived level of pain. Some individuals believe that it is reasonable for postoperative patients to experience moderate pain while others believe that any level of pain should be treated.

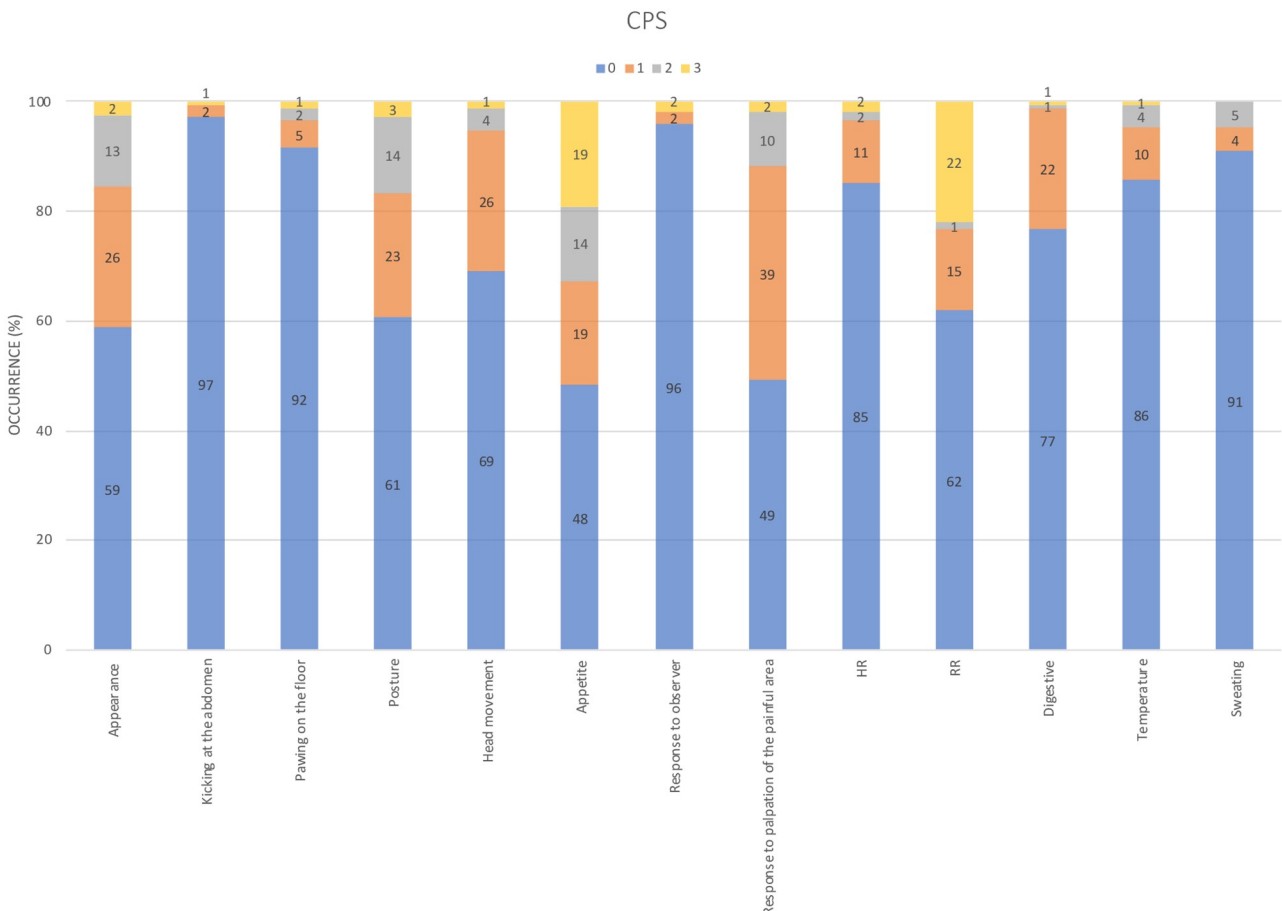

**Fig 5. Distribution of scores for each item of the Composite Pain Scale (CPS) at all perioperative moments.**

Considering that unlike in human patients, where self-reporting of pain is conceivable, there is no gold standard instrument to compare against a new scoring tool in animals; an alternative model is to use a visual analog scale as a 'gold standard' as described in previous studies [7, 11, 12, 21–24] and in children and elderly people [41, 42]. In our case, the criterion validity analysis, aimed at determining the efficacy of the scales based on mutual comparisons, showed a moderate to good correlation among the different scoring tools. An alternative method used to assess criterion validity was to compare the inter-evaluator agreement against the reference evaluator as reported before in cats [22, 25], cattle [21], sheep [24], pigs [23] and children [43]. According to those criteria, the inter-observer agreement was very good only for the lead investigator and ranged from reasonable to good for the other evaluators for both composite scales.

The item-total correlation coefficient provides information regarding the importance of each item to the scale and identifies items that afford a relevant contribution to the total score. This correlation was acceptable for only 50% of the items of the UHAPS (5 out of 10) and 38% of the CPS (5 out of 13). In agreement with the item-total data, the principal component analysis revealed that both scales presented only one dimension and only 50% and 31% of the items of the UHAPS and CPS, respectively were associated in representative dimension. Furthermore, low load values indicated that some items contribute little to both scales, suggesting that some of the evaluation criteria in those scales might be inadequate or obsolete, calling for their

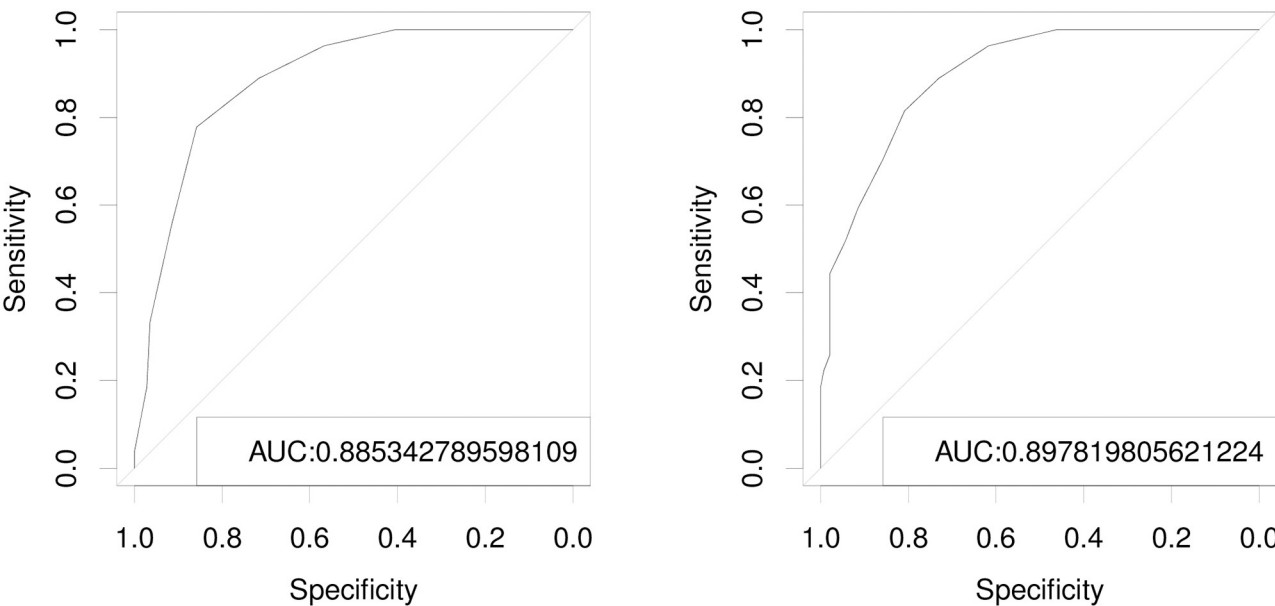

**Fig 6. ROC curve and area under the curve (AUC) for the Unesp-Botucatu horse acute pain scale (UHAPS) (left) and the Composite Pain Scale (CPS) (right).** *ROC: Receiver operating characteristic curve; AUC 1 excellent; AUC 0.9–0.8 very good; AUC 0.8–0.7 good; AUC 0.7–0.6 sufficient; AUC 0.6–0.5 bad; AUC < 0.5 not-discriminative.

**Table 10. Youden index based on highest sensitivity and specificity for the Unesp-Botucatu horse acute pain scale (UHAPS), the Composite Pain Scale (CPS), and unidimensional scales.**

| Scales | Scores of Youden index | Specificity % | Sensitivity (%) | Youden Index |
|---|---|---|---|---|
| CPS | 7 (0–39) | 81 | 81 | 62 |
| UHAPS | 5 (0–17) | 86 | 78 | 64 |
| Visual analog scale | 6 (0–10 cm) | 96 | 96 | 92 |
| Simple descriptive | 3 (0–5) | 95 | 96 | 91 |
| Simple numeric | 5 (0–10) | 94 | 96 | 90 |

UHAPS—Unesp-Botucatu horse acute pain scale; CPS–composite pain scale; Youden index = (Specificity + Sensitivity) - 1

**Table 11. Number of horses requiring rescue analgesia would be indicated based on clinical experience and the Youden index.**

| Moments | Experience | UHAPS Score ≥5 | CPS Score ≥7 | Simple numeric Score ≥5 | Simple descriptive Score ≥3 | Visual analog scale ≥6 |
|---|---|---|---|---|---|---|
| M0 | 3 (7%) | 4 (10%) | 9 (21%) | 5 (12%) | 5 (12%) | 5 (12%) |
| M1 | 10 (24%) | 9 (21%) | 12 (29%) | 10 (24%) | 9 (21%) | 9 (21%) |
| M2 | 9 (21%) | 18 (43%) | 16 (38%) | 11 (26%) | 11 (26%) | 11 (26%) |
| M3 | 5 (12%) | 10 (24%) | 12 (29%) | 8 (19%) | 8 (19%) | 7 (17%) |
| Total | 27 | 41 | 49 | 34 | 33 | 32 |

Unesp-Botucatu horse acute pain scale (UHAPS); Composite pain scale (CPS); Visual analog scale.

exclusion from the rating system in future refinements prior to retesting them again in clinical patients.

The internal consistency was close but below 0.6 for most of the items in both scales, indicating that the ratio of scale items to each other is minimally acceptable. The low internal consistency is probably related to the limited sensitivity of the scale items. When interpreting Cronbach's α coefficient, it should be considered that, firstly, this index is a characteristic of the test score, not the test as a whole [37], that is, it depends on both the population tested and the test itself. A high α value is a prerequisite for internal consistency but does not guarantee it. Multidimensional scales with many items tend to have higher α values, which are not always desirable [37]. Comparing the results of this study with other studies in other species, the need for refinement of both composite scales is apparent. For example, in other studies examining the composite pain scales in cats [22], cattle [21] and pigs [23], Cronbach's internal consistencies were equal or above 0.84, and thus considered excellent. Results of internal consistency followed the same pattern of the principal component analysis and item-total correlation, strengthening the argument that these items (looking at the flank, kicking at the abdomen, lifting hind limbs, pawing on the floor and heart rate for UHAPS and kicking at the abdomen, response to observer, response to palpation of the painful area, heart and respiratory rates, digestive sounds, temperature and sweating for the CPS) should be excluded from the scales because they do not contribute to the construct. The unsatisfactory results for looking at the flank and lifting hind limbs are somewhat surprising. In a recent study, their frequency actually increased in horses with postorchiectomy pain, and they were unaffected by the time of day, anesthesia regimen, and analgesia. Otherwise in accordance with our results, reduction in time spent eating (equivalent to the appetite for CPS) and looking at the back of the stall (equivalent to positioning in the stall for UHAPS), and decreased walking (equivalent to locomotion for UHAPS) were the other behaviors associated only with pain [20].

Sensitivity and specificity are commonly used tests for laboratory diagnostic purposes [19]; however, pain is not a binary response. Specificity was good for the UHAPS and moderate for CPS, so these instruments correctly diagnose pain-free horses in most cases. Still, neither composite scale was sensitive, so they fail to correctly reveal whether horses experience pain. In previous studies examining the value of pain scales in other species and horses [12, 21–25], the animals were submitted to elective surgeries and were pain-free preoperatively, different from this study. For that reason, only pain-free horses (at M0) were considered when calculating specificity (total scores of UHAPS < 5 and CPS < 7). Sensitivity was, nevertheless, weak with both scales, either because M1 and M2 did not necessarily represent periods of intense pain due to the variety of perioperative analgesic protocols used or because none of these scales is sensitive enough, a point that requires further studies.

Responsiveness is 'the ability of an instrument to measure a meaningful or clinically important change in a clinical state' [44]. It is either considered a third measurement property of an instrument/tool together with reliability and validity or a construction validity test [19]. All scales tested in this study demonstrated responsiveness with pain scores higher postsurgery than presurgery, for all observers for UHAPS and for all evaluators except the veterinary student for the unidimensional scales and for four of six observers for the CPS. Likewise, the number of horses judged as needing rescue analgesia according to the Youden index progressively increased from M0 to M2 for both composite scales. However, the low sensitivity of the scales reflects the low pain scores postsurgery (below the cut-off point) and in the small percentage of horses requiring rescue analgesia according to the cut-off point determined on the basis of the Youden index.

The postsurgery pain scores in this study are similar to those reported previously in horses undergoing orthopedic surgery [9], however in our patient cohort, postoperative analgesia

with NSAIDs did not significantly reduce pain scores as would be anticipated based on study design used and a number of references [12, 21–25]. Otherwise pain scores at M2 remained unexpectedly high, either because the analgesic treatment was not effective enough to suppress pain expression at the time the video-clip was recorded, given that neither time of treatment (ordered in a dose decided by the primary clinician) nor the time of video-recordings were matched or because the choice of postoperative analgesia was oral phenylbutazone in most horses. Surprisingly, a shift was observed at the time of greatest pain from M1 (prior to postoperative analgesia) to M2 (1 to 3 hours after analgesic drug treatment), which was approximately 10 hours postsurgery. The fact that horses were offered hay *ad libitum* after surgery possibly affected the bioavailability of oral phenylbutazone, which depends on food availability. Hhay given to horses near the time of medication prolongs intestinal drug absorption for at least 10 hours with a maximum time to peak levels (Tmax) increasing to 12 hours [45, 46]. Thus, the period evaluated as 'most painful' ranged from M1 to M2.

An important limitation of all scales tested was that at the moment of supposedly maximum pain (M1), the median scores were 13% for CPS, 18% for UHAPS and about 25% for the unidimensional scales of the maximum possible scores. Watching the video clips in a random order of horses and moments made 5 of the 6 observers blinds to the time points, even though certain clues, such as the presence of IV catheters and leg bandages or similar, allowed the attentive observer to at least distinguish between pre and postoperative recordings. The differences in reproducibility among evaluators and limited responsiveness of the scales in this study might be related to the different methodological design from that used in previous studies. Although like in this one, in the original studies that developed the CPS [8] and the UHAPS [12] the evaluators were blinds, the differences were that those studies were performed under controlled experimental conditions by using only one standard nociceptive stimulus (i.e. amphotericin-B intra-articular injection for CPS and orchiectomy for the UHAPS), unlike here horses were pain-free at the baseline moment; only moments, rather than horses and moments (like here) were randomized, and evaluators were not allowed to rewind and re-review the videos. There are 'pros' and 'cons' in this approach. The 'cons' is that that the observers might miss some details/behaviors that could be better assessed or picked up by reviewing the videos and 'pros' was that a single video viewing would guarantee more uniformity in assessment among all evaluators. Pain assessment within veterinary hospitals is typically in real-time without video recordings that could be replayed.

The present study tried to mimic a clinical situation in which there is little time to evaluate the animals. In the study by Van Loon et al. [9], responsiveness was satisfactory even in a clinical setting, possibly due to a larger number of evaluated animals, the fact that the evaluators were not blinded to the conditions of the animals; pain was assessed in real time and during a longer (i.e. 10-minute) observation period, compared to the short length of video clips in this study, even considering that lead investigator carefully tried to condense all behaviors displayed in the 15-minute recordings. Those factors might have significantly influenced the evaluators' judgments and contributed to the assignment of higher pain scores in the immediate postoperative period on that study [9]. In the present study, there was less bias based on the expectation for five blind observers. Another factor that may have curtailed responsiveness was the fact that 32 of the 42 animals underwent orthopedic surgery (S1 Table) and thus most likely suffered some degree of pain prior to surgery. The important point is that the sensitivity to change is not only a characteristic inherent of an instrument but it is also related to the effects of an intervention [19]. Therefore, the low percent of change in pain scores observed in the current study also reflected the impact of the intervention, given the variety of procedures and intensities of pre-operative pain, and variable response to postoperative analgesia.

When analyzing the reference evaluators' ratings individually, prior to surgery four and nine horses already had a score calling for rescue analgesia according to the UHAPS and the CPS scales, respectively. In addition, on-site evaluations or distraction of the horses by people and/or other animals passing through the barn may disrupt ongoing discomfort behavior in hospitalized horses, thereby compromising the pain rating and detecting responsiveness. Simultaneously to the current study, a parallel study was performed, in which 20 of the clinical patients were video-recorded for 24 hours, which included a caretaker visit (either to observe and examine or to administer treatment) that was both preceded and followed by one hour of no disturbance (no one interacting with the horse and no indication from the video of the presence of staff or other disturbance in the barn) [3]. This study revealed immediate changes in discomfort behavior in horses as soon as an observer or caretaker approached the animal or entered its stall. Consequently, even in moments of severe pain, horses may not have displayed certain pain-related behaviors as a self-protective strategy [3]. This emphasizes the importance of monitoring animals when they are undisturbed by using remote recording modalities [12, 20]. To allow for undisturbed pain/discomfort behavior monitoring, one should consider i) omitting any recording of physiological parameters and hence eliminating those items from the composite pain scales, or ii) obtaining them at a different time than remote observation, or iii) obtaining those data by means of telemetry technology, such as heart rate monitors. Still, the importance of physiological parameters in assessing pain in animals is disputable [47, 48], and therefore they may all be eliminated after refinement. Another point to consider is that after surgery, all animals were under the effect of perioperative analgesics, which probably decreased or even suppressed pain and reduced pain scores.

The maximum total scores reached 9 out of 17 in the UHAPS and 18 out of 39 in the CPS, corresponding to 53% and 46% of the maximum possible scores. These scales include some mutually exclusive items; most of them are probably unspecific (e.g. 'pawing', 'head movements', 'locomotion'), while at least 'kicking at abdomen' is specific for abdominal pain; however only one item contributes little to the total score. Based on the scales published to date, except for abdominal surgery pain, the maximum pain scores are low even in horses with severe pain, i.e., at most 60% in relation to the total sum of the scales [8, 10, 15]. In none of the previous CPS studies [8, 9, 49] scores ever reached this maximum, including horses not previously treated with analgesics [8] and even if the patient had to be euthanized due to unrelenting pain caused by hoof cancer. Trauma patients reached a maximum of 22 of 39 [9]. Considering the distribution of scores for each item recorded with the composite scales to rate perioperative pain (time-pooled data) (Figs 4 and 5), except for 'positioning', all score 2 items for the UHAPS seem redundant, suggesting that the scores should only be binary (0 and 1). For the same reason CPS score levels could also be reduced to three instead of four. Therefore, high scores indicative of intense pain would be closer to the total score.

A well-defined cut-off point defined by the Youden index was obtained for all scales, with an area under the curve between 0.89 and 1. These results agree with the study of Van Loon *et al.* [9] who suggested scores between 5 and 8 as signifying mild pain. Rescue analgesia scores derived from ROC curves represented 29% (5/17) and 18% (7/39) of maximally possible scores for the UHAPS and CPS scales, respectively, compared to 27% of maximally possible scores in cats [22] and 41% in cattle [21] and 33% in pigs [23] and sheep [24], slightly higher than those determined for the CPS. Interestingly, corresponding rescue analgesia scores were equal to 60% of the maximum visual analog scale and simple descriptive scores and to 50% of the maximum simple numeric score, thus well above values determined for the UHAPS and CPS and for the composite and unidimensional scales in other species [22–24]. At the moment of most intense pain (M2), the number of rescue analgesia indication according to Youden index of unidimensional scales was, therefore, lower than that indicated for composite scales showing

that unidimensional scales may be even less sensitive to recognize pain. Those cut-off points shall serve the equine practitioner and clinician as a guide in prescribing analgesic treatment, as they ensure with greater certainty that an animal in significant pain receives proper analgesic treatment (sensitivity). In contrast, an animal not in pain will not receive such therapy (specificity).

The present study had limitations. Considering that evaluators already made a subjective assessment of whether the animal needed analgesia before performing any scoring, the assessors would bias themselves to score patients they deemed in need of analgesia beforehand higher than patients they considered not in need of analgesia. This methodological approach has been used to define the cut-off point for rescue analgesia in cats, cattle, pigs, and sheep [21–25] and was necessary to build the ROC curve and define the Youden index. Regarding the lack of randomization for assigning scoring systems, the unidimensional scales were scored first because if the composite scales were employed first, their descriptive levels indicating pain behaviors might have influenced the assessment/outcome with the unidimensional scales potentially overestimated their reliability and validity.

A number of factors might have confounded the outcome and thus, interpretation of the data analysis: i. one (the lead investigator) of the 6 evaluators was not blinded and therefore likely biased, ii. the reference evaluator selection was based on an arbitrary basis and iii. on-site evaluations and video recordings did not occur at predefined time points relative to analgesic treatments, thus the impact of analgesic medications during times of pain assessments might have been variable from animal to animal.

A truly validated ('gold standard') method of pain assessment in horses is not available to reliably determine the accuracy and efficacy of an assessment tool in clinical practice. Meanwhile, studies should increase evidence towards validation of a pain assessment scoring system rather than consider the scoring system as validated. These studies should include not only the traditional approach of the three 'Cs' validities (content, construct and criterion), but follow psychometric scoring standards including, for example, test content, response processes, internal structure, relations to other variables, and consequences of testing, to broaden the spectrum of these analyses [50].

In summary, according to our results, the practical outcome we suggest for future studies aiming to validate UHAPS and CPS is to reduce the maximal score of each item of these instruments to 1 and 2, respectively. According to the distribution frequency, the highest scores are not present in most of the items and other behaviors than 'positioning in the stall', 'locomotion', 'locomotion when led by the evaluator', 'response to palpation of the painful area', and 'head movements' for UHAPS and 'appearance', 'pawing on the floor', 'posture', and 'head movement' for CPS should be individually reassessed to investigate their importance, because their results of PCA, item-total correlation, and internal consistency were below the ideal. Except for appetite, the contribution of other physiological data for equine pain assessment should be reinvestigated or changed to remote estimation.

## Conclusions

Reliability and responsiveness of CPS and UHAPS were similar to the unidimensional scales. Both, the UHAPS and the CPS presented overall good or very good repeatability. In contrast, the overall reproducibility was variable. Criterion validity was good (UHAPS) and moderate (CPS), responsiveness occurred to postoperative pain but not to rescue analgesia, both instruments were specific, and rescue analgesia cut-off points were well defined. The UHAPS might be somewhat superior with regard to criterion validity, the association among items, item-total correlation, and responsiveness when compared to the CPS. However, both the UHAPS and

CPS scales are, in their current form, suboptimal instruments for assessing pain in equine patients given the poor association between the items of each scale, minimally acceptable internal consistency, and weak sensitivity. Therefore, both composite scales, except for the definition of intervention analgesia, are not apparently superior to unidimensional scoring systems, when used by experienced observers, until they undergo further refinement to exclude unnecessary items or be replaced by other tools for a more consistent pain and discomfort assessment in horses.

## Supporting information

**S1 Table. Demographic data including sex, breed, age, weight, procedure and perioperative analgesia, and institution of each horse included in this validation study.**
(PDF)

**S2 Table. Anesthetic and analgesic drug protocols.**
(PDF)

**S3 Table. Repeatability of the UHAPS, CPS and unidimensional scales to assess perioperative pain in horses.**
(PDF)

**S4 Table. Reproducibility of the UHAPS, CPS and unidimensional scales to assess perioperative pain in horses.**
(PDF)

**S5 Table. Median (range) scores of the UHAPS and CPS and unidimensional scales in the perioperative period in horses.**
(PDF)

**S1 Dataset. Data equine pain scales.**
(XLSX)

## Acknowledgments

The authors would like to thank Jaime Miller, CVT, for participating in the remote video evaluations; Dr. Juliana Alonso, surgeons Dr. João Pedro Pfeifer, MSc Gustavo dos Santos Rosa and DVM Heitor Cestari for performing the surgeries and postoperative care, especially during the colic syndrome emergency.

## Author Contributions

**Conceptualization:** Paula Barreto da Rocha, Bernd Driessen, Sue M. McDonnell, Klaus Hopster, Marilda Onghero Taffarel, Stijn Schauvliege, Stelio Pacca Loureiro Luna.

**Data curation:** Paula Barreto da Rocha, Laura Zarucco, Miguel Gozalo-Marcilla, Charlotte Hopster-Iversen, Thamiris Kristine Gonzaga da Rocha.

**Formal analysis:** Pedro Henrique Esteves Trindade, Stelio Pacca Loureiro Luna.

**Funding acquisition:** Bernd Driessen, Stelio Pacca Loureiro Luna.

**Investigation:** Paula Barreto da Rocha, Sue M. McDonnell, Laura Zarucco, Miguel Gozalo-Marcilla, Charlotte Hopster-Iversen, Thamiris Kristine Gonzaga da Rocha, Bruna Bodini Alonso.

**Methodology:** Paula Barreto da Rocha, Bernd Driessen, Sue M. McDonnell, Marilda Onghero Taffarel, Stelio Pacca Loureiro Luna.

**Project administration:** Paula Barreto da Rocha, Bernd Driessen, Sue M. McDonnell, Klaus Hopster, Stijn Schauvliege, Stelio Pacca Loureiro Luna.

**Resources:** Bernd Driessen, Stelio Pacca Loureiro Luna.

**Supervision:** Bernd Driessen, Stelio Pacca Loureiro Luna.

**Validation:** Stelio Pacca Loureiro Luna.

**Visualization:** Paula Barreto da Rocha, Bernd Driessen, Stelio Pacca Loureiro Luna.

**Writing – original draft:** Paula Barreto da Rocha, Bernd Driessen, Sue M. McDonnell, Stijn Schauvliege, Stelio Pacca Loureiro Luna.

**Writing – review & editing:** Paula Barreto da Rocha, Bernd Driessen, Sue M. McDonnell, Stelio Pacca Loureiro Luna.

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
