## [Decision Letter · Decision Letter 0]

5 May 2021

PONE-D-21-04051

A critical evaluation for validation of composite and unidimensional postoperative pain scales in horses: a multicenter study

PLOS ONE

Dear Dr. Luna,

Thank you for submitting your manuscript to PLOS ONE. After careful consideration, we feel that it has merit but does not fully meet PLOS ONE’s publication criteria as it currently stands. Therefore, we invite you to submit a revised version of the manuscript that addresses the points raised during the review process.

ACADEMIC EDITOR: The manuscript was revised by experts and it requires major revision before it can be accepted for publication. Please take into consideration the important points raised by reviewer #2 when revising this manuscript. 

We look forward to receiving your revised manuscript.

Kind regards,

Antonio Humberto Hamad Minervino, Ph.D.

Academic Editor

PLOS ONE

Additional Editor Comments (if provided):

The manuscript was revised by experts and it requires a major revision before it can be accepted for publication. Please carefully check the comments made by reviewer #2 and revise the manuscript.

Journal Requirements:

2) Thank you for stating the following in the Competing Interests section:

[The authors have declared that no competing interests exist.].

We note that you received funding from a commercial source: Narkovet Consulting®, LLC,

3) Please include captions for your Supporting Information files at the end of your manuscript, and update any in-text citations to match accordingly. Please see our Supporting Information guidelines for more information: http://journals.plos.org/plosone/s/supporting-information.

4) Your ethics statement should only appear in the Methods section of your manuscript. If your ethics statement is written in any section besides the Methods, please delete it from any other section.

Reviewers' comments:

Reviewer's Responses to Questions

**Comments to the Author**

1. Is the manuscript technically sound, and do the data support the conclusions?

Reviewer #1: Yes

Reviewer #2: Partly

2. Has the statistical analysis been performed appropriately and rigorously? 

Reviewer #1: Yes

Reviewer #2: Yes

3. Have the authors made all data underlying the findings in their manuscript fully available?

Reviewer #1: Yes

Reviewer #2: Yes

4. Is the manuscript presented in an intelligible fashion and written in standard English?

Reviewer #1: Yes

Reviewer #2: Yes

5. Review Comments to the Author

Reviewer #1: The manuscript presents excellent results from the assessment of different pain scales (composite and unidimensional) in the postoperative period of horses.

The study presents a robust statistical evaluation and analyzes criteria that are important to understand the advantages and limitations of the different types of scale.

I suggest adding the p values for the Spearman correlation analysis in tables 3 and 6.

Reviewer #2: The authors are applauded for their efforts to analyze and report this complicated analysis of a large data set. The topic is important to equine medicine/surgery, the experimental design is well-explained, and the information is likely important in our continued search for the best possible way to assess/measure pain in equine patients. It is always challenging to conduct this type of pain assessment in a non-standardized population of clinical patients. I think the authors do a good job in trying to manage the extraneous variables and diminish bias. The manuscript suffers, however, from a lack of focus and clarity. The manuscript is difficult to interpret because of the wide variety of analyses, acronyms, and competing/conflicting statements.

A table of acronyms should be included for reference. A similar table with concise definitions of terminology such as reliability, repeatability, sensitivity, specificity, consistency, validity, etc would be helpful. I realize that descriptors for most terms are included in Table 1 in conjunction with a description of the statistical tests which were utilized, but the definitions are not easily interpreted by scientists who may not be experts in this type of experimental design. I strongly recommend that the manuscript be written with language that makes the information more broadly accessible without sacrificing the precision that is important for interpretation of results.

The stated objective for the study is to compare results obtained with select unidimensional or composite pain scales by evaluators with variable backgrounds. I don't think that the authors can draw any meaningful conclusions about the experience of evaluators based on only 1 of each type of individual. A much larger number of individuals (e.g. many students, many technicians, many surgeons, many internists) would be required to make conclusions regarding whether the experience of the individual performing the assessment has a significant impact on the assessment.

It was interesting that the need for rescue analgesia appeared to have the most intra-observer variability. This type of assessment may depend on many factors other than the perceived level of pain. Some individuals believe that it is reasonable for post-operative patients to experience moderate pain while other individuals believe all pain should be minimized or treated. For example, some individuals may believe that pain should be treated with a VAS of 2 and others may believe that pain should not be treated unless there was a VAS of 8. These are philosophical differences that were not addressed in the discussion. I'm not sure that the need for rescue analgesia is appropriate to be included in the analysis in the same way as the presumably more objective scoring systems. (This doesn't mean that it shouldn't be included in the manuscript but perhaps it should be handled in a different way or at least discussed.)

6. PLOS authors have the option to publish the peer review history of their article (what does this mean?). If published, this will include your full peer review and any attached files.

Reviewer #1: No

Reviewer #2: No

---

## [Author Response · Author response to Decision Letter 0]

25 Jun 2021

Additional Editor Comments (if provided):

The manuscript was revised by experts and it requires a major revision before it can be accepted for publication. Please carefully check the comments made by reviewer #2 and revise the manuscript.

Answer: The authors appreciate the editor and the reviewers for taking the time and effort to review this manuscript. All corrections were performed according to the editor´s and reviewers' suggestions, as described below

Journal Requirements:

Answer: adjustments have been performed to meet PLOS ONE´s style requirements

2) Thank you for stating the following in the Competing Interests section:

[The authors have declared that no competing interests exist.].

We note that you received funding from a commercial source: Narkovet Consulting®, LLC,

Answer: The Competing Interests Statement has been included in the cover letter and manuscript.

3) Please include captions for your Supporting Information files at the end of your manuscript, and update any in-text citations to match accordingly. Please see our Supporting Information guidelines for more information: http://journals.plos.org/plosone/s/supporting-information.

Answer: supporting Information file captions were included, and citations were updated in-text 

4) Your ethics statement should only appear in the Methods section of your manuscript. If your ethics statement is written in any section besides the Methods, please delete it from any other section.

Answer: the ethics statement appears only in Methods section now.

Reviewers' comments:

Reviewer's Responses to Questions

Comments to the Author

1. Is the manuscript technically sound, and do the data support the conclusions?

Reviewer #1: Yes

Reviewer #2: Partly

2. Has the statistical analysis been performed appropriately and rigorously?

Reviewer #1: Yes

Reviewer #2: Yes

3. Have the authors made all data underlying the findings in their manuscript fully available?

Reviewer #1: Yes

Reviewer #2: Yes

4. Is the manuscript presented in an intelligible fashion and written in standard English?

Reviewer #1: Yes

Reviewer #2: Yes

5. Review Comments to the Author

Reviewer #1: The manuscript presents excellent results from the assessment of different pain scales (composite and unidimensional) in the postoperative period of horses.

The study presents a robust statistical evaluation and analyzes criteria that are important to understand the advantages and limitations of the different types of scale.

I suggest adding the p values for the Spearman correlation analysis in tables 3 and 6.

Answer: the authors appreciate your time and effort in reviewing this manuscript and your positive comments. 

The p values for the Spearman correlation analysis were included in tables 3 and 6 as required.

Reviewer #2: The authors are applauded for their efforts to analyze and report this complicated analysis of a large data set. The topic is important to equine medicine/surgery, the experimental design is well-explained, and the information is likely important in our continued search for the best possible way to assess/measure pain in equine patients. It is always challenging to conduct this type of pain assessment in a non-standardized population of clinical patients. I think the authors do a good job in trying to manage the extraneous variables and diminish bias. The manuscript suffers, however, from a lack of focus and clarity. The manuscript is difficult to interpret because of the wide variety of analyses, acronyms, and competing/conflicting statements.

Answer: the authors appreciate your time and effort in reviewing this manuscript and your constructive comments. We did our best to improve clarity. Although an abbreviation list was already available in the previous version, we excluded the acronyms in the text to provide an easier reading. 

A table of acronyms should be included for reference. A similar table with concise definitions of terminology such as reliability, repeatability, sensitivity, specificity, consistency, validity, etc would be helpful. I realize that descriptors for most terms are included in Table 1 in conjunction with a description of the statistical tests which were utilized, but the definitions are not easily interpreted by scientists who may not be experts in this type of experimental design. I strongly recommend that the manuscript be written with language that makes the information more broadly accessible without sacrificing the precision that is important for interpretation of results.

Answer: concise definitions of terminology and the rationale behind the methods used for statistical analysis has been included as a new column in Table 1. Results are better explained in lines 301-303, 339-348, 351-354 and, 392-395. Some comments have also been included in Discussion (lines 499-511,514-515, and 530-532). We hope the manuscript is more comprehensible and easier to follow.

The stated objective for the study is to compare results obtained with select unidimensional or composite pain scales by evaluators with variable backgrounds. I don't think that the authors can draw any meaningful conclusions about the experience of evaluators based on only 1 of each type of individual. A much larger number of individuals (e.g. many students, many technicians, many surgeons, many internists) would be required to make conclusions regarding whether the experience of the individual performing the assessment has a significant impact on the assessment.

Answer: the authors agree with your comment. It would be necessary at least six individuals of each category to draw any conclusion about the results of the observers. All points regarding this topic were excluded from Discussion.

It was interesting that the need for rescue analgesia appeared to have the most intra-observer variability. This type of assessment may depend on many factors other than the perceived level of pain. Some individuals believe that it is reasonable for post-operative patients to experience moderate pain while other individuals believe all pain should be minimized or treated. For example, some individuals may believe that pain should be treated with a VAS of 2 and others may believe that pain should not be treated unless there was a VAS of 8. These are philosophical differences that were not addressed in the discussion. I'm not sure that the need for rescue analgesia is appropriate to be included in the analysis in the same way as the presumably more objective scoring systems. (This doesn't mean that it shouldn't be included in the manuscript but perhaps it should be handled in a different way or at least discussed.)

Answer: Thank you for your comment. This was included in Discussion (lines 465-469). According to your suggestion, rescue analgesia results were excluded from Tables 2 and 4 of the main text, as the same information is provided in Supporting Information (S3 and S4 Tables).

---

## [Decision Letter · Decision Letter 1]

21 Jul 2021

A critical evaluation for validation of composite and unidimensional post-operative pain scales in horses

PONE-D-21-04051R1

Dear Dr. Luna,

We’re pleased to inform you that your manuscript has been judged scientifically suitable for publication and will be formally accepted for publication once it meets all outstanding technical requirements.

Kind regards,

Antonio Humberto Hamad Minervino, Ph.D.

Academic Editor

PLOS ONE

Additional Editor Comments (optional):

I am pleasure to inform that this manuscript can be publish in the journal. The authors successfully correct all the issues raised during review and the manuscript meets all the criteria for publication at PLoS One.

Reviewers' comments:

Reviewer's Responses to Questions

**Comments to the Author**

1. If the authors have adequately addressed your comments raised in a previous round of review and you feel that this manuscript is now acceptable for publication, you may indicate that here to bypass the “Comments to the Author” section, enter your conflict of interest statement in the “Confidential to Editor” section, and submit your "Accept" recommendation.

Reviewer #2: All comments have been addressed

2. Is the manuscript technically sound, and do the data support the conclusions?

Reviewer #2: (No Response)

3. Has the statistical analysis been performed appropriately and rigorously? 

Reviewer #2: (No Response)

4. Have the authors made all data underlying the findings in their manuscript fully available?

Reviewer #2: (No Response)

5. Is the manuscript presented in an intelligible fashion and written in standard English?

Reviewer #2: (No Response)

6. Review Comments to the Author

Reviewer #2: (No Response)

7. PLOS authors have the option to publish the peer review history of their article (what does this mean?). If published, this will include your full peer review and any attached files.

Reviewer #2: No

---

## [Editor Report · Acceptance letter]

27 Jul 2021

PONE-D-21-04051R1 

A critical evaluation for validation of composite and unidimensional postoperative pain scales in horses 

Dear Dr. Luna:

I'm pleased to inform you that your manuscript has been deemed suitable for publication in PLOS ONE. Congratulations! Your manuscript is now with our production department. 

Kind regards, 

on behalf of

Dr. Antonio Humberto Hamad Minervino 

Academic Editor

PLOS ONE